# The Self-Stranding Behavior of a Killer Whale (*Orcinus orca*) in Inner Danish Waters and Considerations concerning Human Interference in Live Strandings

**DOI:** 10.3390/ani13121948

**Published:** 2023-06-10

**Authors:** Aage Kristian Olsen Alstrup, Charlotte Bie Thøstesen, Kirstin Anderson Hansen, Christian Sonne, Carl Christian Kinze, Lars Mikkelsen, Annika Thomsen, Peter Povlsen, Hanne Lyngholm Larsen, Anne Cathrine Linder, Sussie Pagh

**Affiliations:** 1Department of Nuclear Medicine and PET, Aarhus University Hospital, 8200 Aarhus, Denmark; 2Department for Clinical Medicine, Aarhus University, 8200 Aarhus, Denmark; 3Fisheries and Maritime Museum, 6710 Esbjerg, Denmark; cbt@fimus.dk; 4Fjord & Bælt, 5300 Kerteminde, Denmark; kirstin@fjordbaelt.dk; 5Department of Biology, University of Southern Denmark, 5000 Odense, Denmark; 6Department of Ecoscience, Aarhus University, 4000 Roskilde, Denmark; cs@ecos.au.dk; 7Cetacean Atlas of Denmark, 1870 Frederiksberg, Denmark; cck@hvaler.dk; 8Lars Mik Photography, 7560 Hjerm, Denmark; lars@webareus.dk; 9The North Sea Oceanarium, 9850 Hirtshals, Denmark; at@nordsoemail.dk; 10Department of Chemistry and Bioscience, Faculty of Engineering and Science, Aalborg University, 9220 Aalborg, Denmark; ppovls18@student.aau.dk (P.P.); hannell@bio.aau.dk (H.L.L.); sup@bio.aau.dk (S.P.); 11National Institute of Aquatic Resources, Technical University of Denmark, 2800 Kongens Lyngby, Denmark; acali@aqua.dtu.dk

**Keywords:** behavior, euthanasia, killer whale, life stranding, rescue operation, strandings

## Abstract

**Simple Summary:**

When live whales strand, we need to choose our actions about whether the whale should be rescued, euthanized for animal welfare reasons, or left alone without any kind of interference whatsoever. This important choice is based on sparse information about its true health condition, since this is often either not available or difficult to assess. Based on the stranding of a live male killer whale in inner Danish Waters in 2021 and 2022, we discuss here these dilemmas and how to decide upon our action given these knowledge gaps, in order to predict the whale’s chances of survival.

**Abstract:**

The rescue attempts of stranded whales and euthanasia considerations must include condition assessments of the individual involved, but this is challenged by our insufficient knowledge about the health statuses of these whales. Here, we describe three separate strandings of a young male killer whale (*Orcinus orca*) in shallow Danish waters during 2021–2022. During the first two stranding events, the whale exhibited remarkable behavior and, after refloating attempts and several kilometers of swimming, it returned to shallow water. This suggests that it actively chose to be in this shallow water, perhaps to ensure free airways and respiration. During the last stranding, it stayed in shallow water for 30 days, during which, euthanasia was considered due to its seemingly worsened condition, including a collapsed dorsal fin. However, suddenly, the whale swam away and, a year later, he was seen alive, confirming that euthanasia would have been the wrong decision. This case raises an important question as to when and under what circumstances active human interventions, such as refloating attempts, should be launched and when euthanasia should be carried out. Every stranding is unique and decisions should be based on thorough considerations of the animal’s health and the chance of a successful rescue.

## 1. Introduction

Small groups or single killer whales (*Orcinus orca*) are regularly observed in the Danish waters of both the North Sea and Skagerak [1]. Killer whales have been documented in Danish and adjacent waters since 1545 [2,3]. The earliest Danish find was a 19” female killed on 27 December 1679 (Julian date) in the Randers Fjord [4], which was depicted in a reprinted oil painting [5].

In the period of 1700–1799, four documentations of killer whales in Denmark were recorded. All four events were directed catches for consumption, involving a total of 19 individuals. In the years of 1800–1899, twenty events were documented, involving 33 individuals, 11 of which were singletons. For the 1900s, the corresponding figures were 19 individuals that were all singletons. These finds include similar and even the same localities, with comparable circumstances to the present event described and lonely male individuals, too.

In 1960, all cetacean species occurring in Danish waters, except the harbor porpoise (*Phocoena phocoena*), became protected by law, leading to the end of directed catches. Up until 1960, live stranded individuals were considered a natural resource, and were thus killed at the stranding site.

On 2 June 1980, the first Danish rescue attempt was launched in connection with the live stranding event of a killer whale. A 322 cm female was transported from the Skagerrak coast to Aalborg Zoo, where it died the following day. According to the necropsy report, “a weak heart” eventually caused the death. A decade later—in the summer of 1990—another young killer whale (a 395 cm male) was found dead in the Randers Fjord, where it had resided for several weeks, during which, it had drawn the interest of thousands of people. A rescue operation to guide the whale back to the open Kattegat was unsuccessful. The whale exhibited signs of starvation and this was possibly the main cause of its death (Extracts from the Danish Whale Archive curated by Carl C. Kinze). Between 1990–2009, a total of four dead killer whales have been necropsied (Figure 1A).

Here, we describe the case of a young male killer whale that live stranded in shallow waters on three separate occasions, once in 2021 and twice in 2022. In particular, the strandings in 2022 gave rise to reflections due to its remarkable behavior with these repeated strandings, which has not previously been described. Furthermore, it raised the question about under what circumstances active human interventions, such as refloating, should be launched and when euthanasia should be considered.

## 2. Materials and Methods

During the three strandings of the young killer whale in 2021 and 2022, daily observations were conducted (counting its respiration rate, dorsal fin position, and whether there were signs of skin lesions) using manual inspections or with a drone. These observations were supplemented with an inspection by a veterinarian (AKOA) during the second stranding. This inspection focused on whether a refloating attempt could be justifiable (focusing on the whale’s response to the rescue team, its position in the water and ability to actively lift its blowhole above the surface of the water, its respiration rate, and the presence of any external lesions under or over the water line, as well as the position of its dorsal fin). The killer whale was identified by comparing the outlines of the saddle patch and minor scars on the photo ID recorded. The approximate length of the killer whale was calculated based on simple trigonometry. The photos presented were taken with handheld cameras and drones. All decisions about refloating attempts or euthanasia were made by the Danish authorities (The National Contingency Plan Concerning Strandings of Marine Mammals in Denmark).

## 3. Results

### 3.1. The First Stranding November 2021

The killer whale was stranded off the coast of Als Odde (coordinate 56.70543; 10.33114) in Denmark (Figure 1B) on 18 November 2021, several hundred meters from deep water. Based on its body size (approximately 6.5 m, range: 6.3–6.7 m) and the height of its dorsal fin, it was estimated to be a young male (Figure 2). It was without any visible external skin lesions, had an upright dorsal fin, and seemed to be in a relatively good physical condition, though it was thinner than expected in late autumn. The water depth also seemed to be deep enough for the whale to be able to swim away if it chose to. The Danish authorities decided not to intervene, but to continue with daily observations. The whale laid passively and was motionless in the water, but after four days, on 22 November 2021, it suddenly departed during windy conditions (www.dmi.dk, accessed on 20 January 2023). The wind may have brought more water into the Fjord, making it easier for the whale to leave.

### 3.2. The Second Stranding April 2022

On 8 April 2022, the same individual stranded in shallow waters in Limfjord, near Hals (coordinates 56.99231; 10.29269) (Figure 1B and Figure 3).

The whale was stranded in shallow water (approximately 1.5 m). It seemed to be in a good physical condition. The stranding location was suitable for a refloating attempt because it was situated close to deep water. Therefore, on 10 April 2022, the Danish authorities decided to send a rescue team composed of biologists, a veterinarian, and divers to try to refloat the whale into deeper water. During this refloating, the whale was calm. The veterinarian performed a clinical inspection, while the divers searched the seabed under and around the whale to see if there were any sharp objects that could injure the whale during the refloating attempt. The inspection showed that the whale was able to lift itself up onto its pectoral fins when breathing (stable frequency: 1/90 s). It had an upright dorsal fin, though it was slightly bent to the left side, and it clearly responded to the team’s presence with its body and eye movements, while actively producing whistles and echolocation clicks. There were no signs of external lesions above or below the waterline. It was, therefore, justifiable to try to refloat the whale. The first trial of relocating the whale consisted of a manual attempt by a team of nine people, which tried to push the whale towards the deeper water passage (Figure 4A), hoping that it would commence swimming. However, the muddy seabed and weight of the whale made this impossible. A second attempt was made by two divers and a rescue boat. The divers placed a strap behind the pectoral fins (Figure 4B) to pull the whale into deeper waters (Figure 4C). The first swimming movements of the whale were observed while the pulling was underway. Once in deeper water, it laid still and the strap was removed. After approximately an hour, it started swimming towards Hals and towards Aalborg (Figure 1B), alternately, while showing hunting behavior in the fjord. The next morning, on 11 April 2022, it was found swimming around in The Greenland Harbor (coordinates 57.04821; 10.03684) near Aalborg, about 20 km west of the fjord, where it was rescued (Figure 1B).

### 3.3. The Third Stranding April 2022

On 11 April 2022, the killer whale was observed lying in shallow water in the Nørre Uttrup Marina (Figure 1B), where it stayed for the next 30 days. The whale was monitored daily from a small boat or the shore by the Danish Nature Agency. On 19 April 2022, the whale entered near-shore shallower water and a trained behavioral biologist examined the whale. The whale, floating in a water depth of 2.5 m, was actively clicking and whistling during the physical inspection, but no performed movements were observed. It was not physically obstructed by anything under or above the water. It supported itself by placing its pectoral fins and flukes in the sand. It appeared to be in good condition. The dorsal fin was collapsed and its breathing was strong, clear, and odorless. Due to the water depth, it would have been possible for it to swim away with little effort. Its respiratory rate was 1/15 s on 11–12 April 2022, but dropped to between 1/30 and 1/60 s for the rest of the period, until 8 May 2022. The condition of the whale gradually worsened—showing signs of increased emaciation, dehydration, and skin lesions, and on 9 May 2022, gulls began to peck around its blowhole and hanging dorsal fin, with no visible reaction from the whale (Figure 5). The whale was markedly thinner compared to when it stranded almost a month earlier. During these 30 days, little bodily movement was observed and the whale made no effort to swim to deeper waters or even adjust its position. Therefore, it was quite unexpected that the whale suddenly swam towards the estuary of the fjord on the morning of 11 May 2022. Later the same day, the Danish Nature Agency observed the whale swimming briskly away, slapping with both its pectoral fins and flukes and with its dorsal fin upright, about 2.5 km from the stranding location. This was the last observation of the killer whale in 2022.

On 21 April 2023, the killer whale was once again recorded, around 10 km from the coast of Västra Götaland, Sweden (coordinates 58.038476; 11.291658). It was still alone.

## 4. Discussion

This case raises the question of when active human interventions, such as refloating or euthanasia, should be initiated. Within a day after the rescue operation on 10 April 2022 and after several kilometers of swimming, the whale was again back in shallow water. On 9 May 2022, after 30 days without food and water, very minimal body movement, and attacks from herring gulls, euthanasia was considered, but two days later, the whale swam away without assistance. Within a few hours, it was observed splashing and rolling in deeper water with its dorsal fin upright. We were surprised that its condition could improve so quickly. This case gives us reason for reflection upon when stranded whales should be refloated, euthanized, or simply left alone.

### 4.1. Conservation Status of Killer Whales

The killer whale is listed on the IUCN Red List as “Data Deficient”. The worldwide minimum population of killer whales has been suggested to be 50,000 individuals [6]. However, the morphological, behavioral, dietary, and genetic diversity among different groups or populations of killer whales suggest that separate subpopulations and even subspecies exist [7]. North Atlantic killer whales are recovering following the past harvesting, culling, and live capturing of the species in the 1980s, but new emerging threats, especially chemical pollution, climate change, anthropogenic noise, and lethal interactions with fisheries, may potentially hamper this recovery [8,9]. Considering the size of the population, whether single-stranded individuals survive will probably only be decisive to a limited extent, but it will, of course, contribute to the stranded individual’s well-being.

### 4.2. Earlier Rescue Attempts of Stranded Killer Whales

Unfortunately, not all rescue attempts are successful. Two earlier attempts to rescue killer whales in Denmark in 1980 and 1990 were not successful, which put the focus on this third attempt. The two killer whales in 1980 and 1990 suffered from heart disease and starvation, respectively. Recent rescue attempts of killer whales in Holland and France were apparently unsuccessful, based on non-scientific written news reports [10,11]. Scientists concluded that the killer whale that stranded in the Seine River in France in May 2022 was terminally ill, likely suffering from mucormycosis, a rare fungal infection in whales [10]. The killer whale that stranded in October 2022 off the coast of the Netherlands, near Cadzand, died within few days, after attempts to push the whale back to the Wadden Sea. A veterinarian from the Dutch authorities concluded that the whale was ill and chose to sedate the whale to make its death painless [11]. However, there are also examples of rescue attempts of killer whales with positive outcomes. On 2 May 2017, a group of nine killer whales entered the narrow channel of Trælvikosen, Brønnøysund, off the coast of central Norway [12]. The water depth in the bay was between 1 and 3 m at low tide. While four of the killer whales left the bay on their own after 16 days, a rescue operation was established to guide the five remaining individuals back into the sea. On days 16 and 17, the killer whales in the bay produced loud and repeated calls in association with long bubble streams for several hours at a time, while wandering over the entire bay [12]. The animals seemed to be in a distressed state. To identify potential signs of stress, their behavior and sound recordings were monitored throughout the entrapment and rescue operation; thus, land-based observations from drone imagery and underwater acoustic recordings were combined. The rescue operation took place on 20 May 2017, 19 days after the killer whales entered the bay. Boats and kayaks gathered in a line behind the four whales to move them toward the narrow entrance at high tide [12]. After some time, the whales were directed through the narrow tunnel back into the sea. The entire group of rescued killer whales was resighted the following September, about 700 km north from the entrapment location.

In a review of stranded killer whales in North America, killer whales were found to strand in pursuit of prey; this behavior was otherwise considered rare. These events did not involve a specific sex or age class [13]. Historical records show that many live stranded killer whales have been euthanized or captured for aquariums, where they subsequently died. This review presented five live stranding events since 2002, where all the individuals survived, although one adult male was never seen again [13]. Three adults were stranded on sandy shores and two juveniles were stranded on rocky outcroppings while hunting harbor seals. During three of the live strandings, the stranded individuals were kept cool and wet by human responders, and efforts were made twice to move the animals off the shore. Four individuals rejoined their respective families shortly after being refloated and have been photo-identified on numerous occasions. Moreover, one adult female was pregnant when stranded and gave birth to a healthy calf several months later. The review concluded that the outcomes from these five strandings indicate that human responses are not always necessary, but when they are, they can help preserve the animals’ lives and family bonds [13].

In June 1997, a subadult male killer whale stranded off the coast of New Zealand, where it remained on the beach for around 21 h [14]. The killer whale was observed both before and several times after the stranding and it survived for more than three years after this stranding. At the stranding site, the animal appeared to be uninjured externally, with the exception of a few minor wounds and a blister on its dorsal fin. Refloating was achieved using specially designed “rescue pontoons” (PVC nylon mats) [14].

### 4.3. Considerations on Human Interference in Live Strandings of Whales

Live strandings often prompt heated public discussions about whether the animal is suffering and if the animal should be rescued, sedated, euthanized, or left alone to let nature take its course. Local authorities are often under time pressure to report a clear strategy to the media, addressing how the situation is being dealt with in order to avoid interference from well-intentioned people who want to help the whale.

Before taking any actions related to stranded whales, we suggest the following considerations:(1)The whale’s chance to free itself.(2)Is the stranding location suitable for a refloating attempt without injuring the whale? For example, how far away is deep water and how is the nature of the seabed?(3)The whale’s health condition and its chance of survival post-rescue.(4)A veterinarian’s examination of the whale’s condition.(5)The possible safety risk for personal rescuing or euthanizing the whale.(6)The well-being of the whale before and after the rescue.(7)The well-being of the public and their reaction to the whale’s suffering and a rescue attempt (but never at the expense of the whale’s welfare).

The refloating of the Danish killer whale in 2022 demonstrates the complexity of the issues that authorities are faced with in the occurrence of a live stranding. Whilst no rescue can be guaranteed, refloating attempts at the right time (among other factors, such as an assessment of the whale’s health and bodily conditions) and at a suitable locality may provide the necessary aid for survival. Euthanasia, in the present case, would have been the wrong decision, since the whale eventually swam away and was still alive a year later.

## 5. Conclusions

In conclusion, although the reasons remain obscure, this killer whale appeared to enter shallow water and later leave it of its own volition. This case shows that rescue attempts and euthanasia considerations are difficult. The whale was in good bodily condition and was in relatively deep water, without any obstructions. Every stranding is unique and decisions should be based on thorough considerations of the animal’s health, the chance of a successful rescue, and with reflections upon both the animal’s and people’s well-being. Caution should be taken before deciding to euthanize an animal.

## Figures and Tables

**Figure 1 animals-13-01948-f001:**
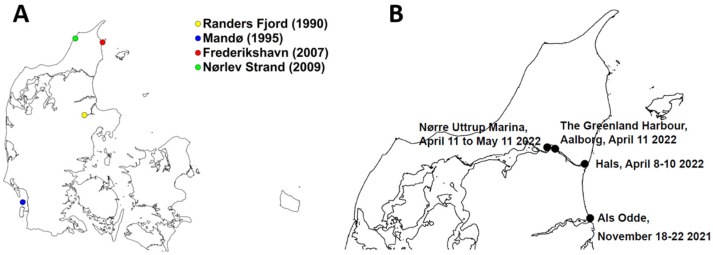
GIS Map (MapInfo Pro v2021) of stranded killer whales (*Orcinus orca*) in Danish waters 1990–2022: Four stranded killer whales in the period 1990–2009, and they were found at the following locations: Randers Fjord (1990), Mandoe (1995), Frederikshavn (2007), and Noerlev Strand (2009) [1,2,3,4,5] (**A**). Locations of the present male killer whale strandings in 2021 and 2022 (**B**).

**Figure 2 animals-13-01948-f002:**
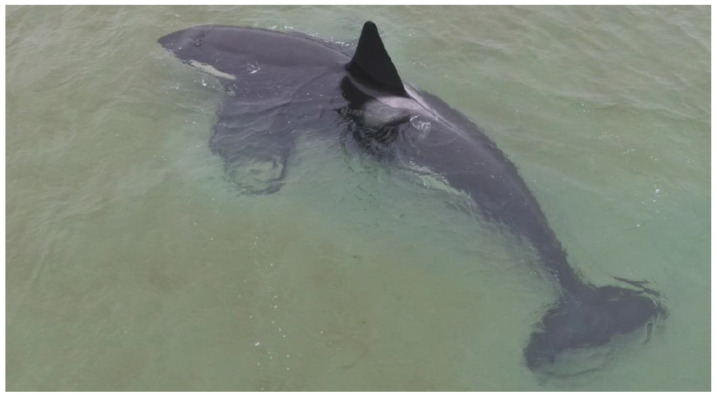
The young male killer whale (*Orcinus orca*; approximately 6.5 m and with a long sword-shaped dorsal fin), beached 18 November 2021, near Als, Denmark.

**Figure 3 animals-13-01948-f003:**
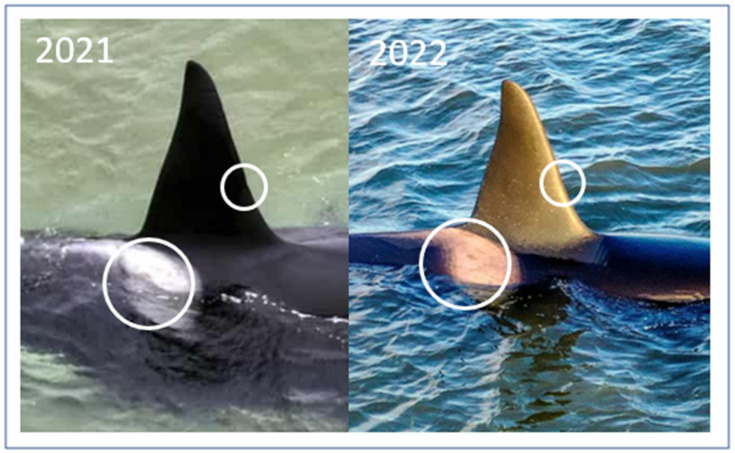
The young male killer whale (*Orcinus orca*) from 2021 and 2022 was identified as the same individual based on photo identification. It was approximately 6.5 m long with a long, sword-shaped dorsal fin, grey saddle patch outline (large circles), and minor dorsal fin scars (small circles).

**Figure 4 animals-13-01948-f004:**
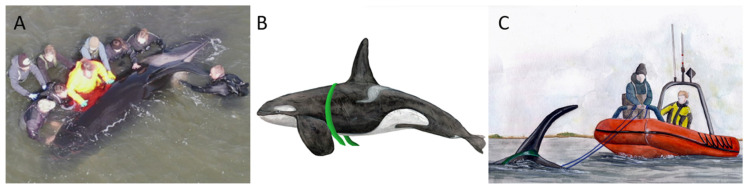
A refloating of the killer whale was carried out on 10 April 2022 by attempting to push the whale into deeper waters (the fairway), but it was too heavy to move (**A**). Subsequently, slings were placed around the whale’s pectoral fins (**B**), and it was then successfully pulled out into deeper waters using an inflatable boat (**C**).

**Figure 5 animals-13-01948-f005:**
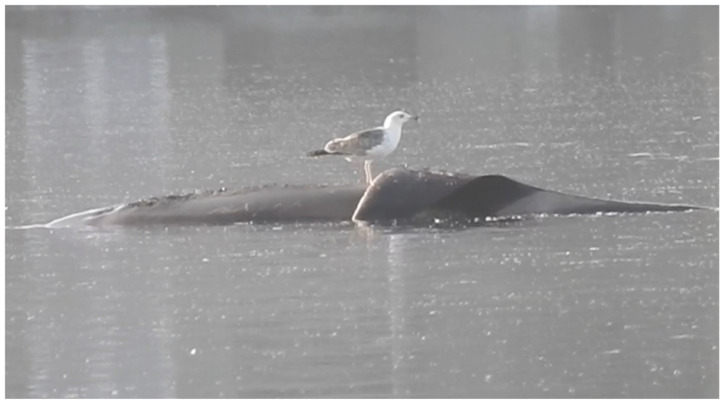
Herring gulls (*Larus argentatus*) were seen sitting on the killer whale on 9 May 2022, pecking at the flat, hanging dorsal fin. The whale reacted to the birds’ presence, but not enough to induce their departure.

## Data Availability

No further data exist.

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
