# Peer review of "The Self-Stranding Behavior of a Killer Whale (Orcinus orca) in Inner Danish Waters and Considerations concerning Human Interference in Live Strandings"

_animals, 2023, doi:10.3390/ani13121948_

Round 1
Reviewer 1 Report
Dear authors, thanks for sharing this case report which is very interesting and useful for the conservation of this species and to gather information on this species.
Please, find below my suggestions to improve the scientific level of your fantastic work.
1. The title on the website is not the same as on the PDF.
2. The title on the PDF should be corrected to: "[...] live strandings".
3. Line 75-76: From where the picture was taken? Aerial survey? Unmanned aerial vehicle?
4. Line 98: Please, it would be interesting to report which "clinical inspection" were performed by the veterinarian.
5. Line 150: It sounds very unusual from an anatomical point of view that the dorsal fin was upright on the same day and after 1 month of inactivity.
6. Lines 177-178: The consideration about euthanasia should be reformulated as no investigations were performed on animal welfare and health status. The fact that the O. orca was alive doesn't show any evidence of the welfare or good health of the animal.
7. Line 181: Please, reformulate the sentence as it is not certain that this is a "unique" case.
8. Lines 183-184: The last sentence is not clear and in contradiction to what reported previously.
Author Response
Reviewer 1:
Dear authors, thanks for sharing this case report which is very interesting and useful for the conservation of this species and to gather information on this species. We thank you for these kind words.
Please, find below my suggestions to improve the scientific level of your fantastic work. We thank you for the constructive suggestions.
- The title on the website is not the same as on the PDF. We have - following the editor's suggestion - now chosen to change the title of the manuscript.
- The title on the PDF should be corrected to: "[...] live strandings". The manuscript, both in word and pdf, is corrected. We followed the recommendation from the editor.
- Line 75-76: From where the picture was taken? Aerial survey? Unmanned aerial vehicle? We apologize for the confusion. A drone was used for this (DJI Mavic 2 Enterprise Advanced (M2EA). We have in the M&M section added where the pictures (photos) were taken from.
- Line 98: Please, it would be interesting to report which "clinical inspection" were performed by the veterinarian. The clinical examination was exclusively based on external observations of the whale and this is also clarified in the manuscript. It was not possible to carry out more in-depth examinations, including paraclinical laboratory examinations, although this would have been both interesting and relevant in order to assess the condition of the whale. Unfortunately, we therefore cannot add any more details about the clinical inspection.
- Line 150: It sounds very unusual from an anatomical point of view that the dorsal fin was upright on the same day and after 1 month of inactivity. Yes, we have also wondered that such big changes could happen in such a short time. Nevertheless, it was what we observed. We have now added in the discussion that we also find this surprising.
- Lines 177-178: The consideration about euthanasia should be reformulated as no investigations were performed on animal welfare and health status. The fact that the O. orcawas alive doesn't show any evidence of the welfare or good health of the animal. We agree with these correct considerations and have therefore revised the text.
- Line 181: Please, reformulate the sentence as it is not certain that this is a "unique" case. Done.
8. Lines 183-184: The last sentence is not clear and in contradiction to what reported previously. Thanks, we have now revised the text in the conclusion.
Reviewer 2 Report
Alstrup et al provide a case study of a killer whale that appears to have stranded itself on purpose and then eventually swam away and survived indicating that euthanizing it would have been a mistake. The paper is well-written and documented and provides an interesting case in a poorly understood but provocative field that is difficult to explore experimentally.
The conclusion should probably say that although reasons remain obscure, the whale appeared to enter shallow water and later leave under its own volition.
Minor points
L 27. Re: chosen to be in shallow water to ensure free airways and respiration. Too strong a statement since we don’t really know the whale’s motivation. Delete the “to” and add which would ensure free….
Fig 1B legend. A map is not observations. Change observations to locations.
L 63. The authors don’t say how they were able to calculate a range in estimated length, and I don’t see it as important. Either delete it or provide more explanation.
L 72. I don’t recognize the abbreviation “sm.” Wind is usually reported in knots? If you are going to speculate on the effect of wind making it easier to leave, at least give the direction in terms of adding to water depth. I think it would be better to mention the wind but not to make potential conclusions.
L 102. Delete “but” when describing dorsal fin.
L 107. Fairway sounds like a golf term and may not be clear to many readers. Are you indicating that it was pointed toward deeper water or a channel?
L 178. Incorrect not false?
Author Response
Reviewer 2:
Alstrup et al provide a case study of a killer whale that appears to have stranded itself on purpose and then eventually swam away and survived indicating that euthanizing it would have been a mistake. The paper is well-written and documented and provides an interesting case in a poorly understood but provocative field that is difficult to explore experimentally. We thank the reviewer for these nice words about the manuscript. Yes, it is a difficult area to investigate experimentally.
The conclusion should probably say that although reasons remain obscure, the whale appeared to enter shallow water and later leave under its own volition. Thanks for the comment. We have now added this to the conclusion in the manuscript.
Minor points
L 27. Re: chosen to be in shallow water to ensure free airways and respiration. Too strong a statement since we don’t really know the whale’s motivation. Delete the “to” and add which would ensure free…. We agree and have rewritten the sentence in accordance with the comment and editor's suggested changes..
Fig 1B legend. A map is not observations. Change observations to locations. Done.
L 63. The authors don’t say how they were able to calculate a range in estimated length, and I don’t see it as important. Either delete it or provide more explanation. It is now deleted.
L 72. I don’t recognize the abbreviation “sm.” Wind is usually reported in knots? If you are going to speculate on the effect of wind making it easier to leave, at least give the direction in terms of adding to water depth. I think it would be better to mention the wind but not to make potential conclusions. We thank you for the comment and have now revised the sentence.
L 102. Delete “but” when describing dorsal fin. Done
L 107. Fairway sounds like a golf term and may not be clear to many readers. Are you indicating that it was pointed toward deeper water or a channel? We thanks, and we now use the words: “deeper water passage”.
L 178. Incorrect not false? We apologize that we do not understand this comment. We haven't written "false" in the latest version of the manuscript?